# Autoimmune Epithelitis and Chronic Inflammation in Sjögren’s Syndrome-Related Dry Eye Disease

**DOI:** 10.3390/ijms222111820

**Published:** 2021-10-30

**Authors:** Yoko Ogawa, Tsutomu Takeuchi, Kazuo Tsubota

**Affiliations:** 1Department of Ophthalmology, Keio University School of Medicine, 35 Shinanomachi, Shinjuku, Tokyo 160-8582, Japan; tsubota@z3.keio.jp; 2Department of Internal Medicine, Division of Rheumatology, Keio University School of Medicine, 35 Shinanomachi, Shinjuku, Tokyo 160-8582, Japan; tsutake@z5.keio.jp

**Keywords:** Sjögren’s syndrome, dry eye, autoimmune epithelitis, chronic inflammation

## Abstract

Autoimmune epithelitis and chronic inflammation are one of the characteristic features of the immune pathogenesis of Sjögren’s syndrome (SS)-related dry eye disease. Autoimmune epithelitis can cause the dysfunction of the excretion of tear fluid and mucin from the lacrimal glands and conjunctival epithelia and meibum from the meibomian glands. The lacrimal gland and conjunctival epithelia express major histocompatibility complex class II or human leukocyte antigen-DR and costimulatory molecules, acting as nonprofessional antigen-presenting cells for T cell and B cell activation in SS. Ocular surface epithelium dysfunction can lead to dry eye disease in SS. Considering the mechanisms underlying SS-related dry eye disease, this review highlights autoimmune epithelitis of the ocular surface, chronic inflammation, and several other molecules in the tear film, cornea, conjunctiva, lacrimal glands, and meibomian glands that represent potential targets in the treatment of SS-related dry eye disease.

## 1. Introduction

Sjögren’s syndrome (SS) is an autoimmune disorder characterized by chronic lymphoplasmacytic infiltration of mainly the lacrimal glands, and salivary glands, leading to dry eye disease, and dry mouth [1,2]. There are several risk factors for SS, including genetic factors under specific epigenetic factors, hormonal imbalances, sex factors, and environmental factors, including viral infection [3]. SS is a female-predominant disease, but it can affect males at a ratio of 1:17 in Japan [4] and 1:10 worldwide [5]. Dysfunction of epithelial cells [6], mesenchymal stroma cells, and immune inflammatory cells including dendritic cells, macrophages, natural killer cells, T cells, B cells, and production of antinuclear autoantibodies plays key roles of the development of this disease [5,7].

Surface barrier immunity on the ocular surface involves the communication between local and systemic inflammatory cells and interstitial mesenchymal cells or epithelial cells that interact with specific cell surface receptors and soluble factors to maintain the homeostasis of the ocular surface through innate and adaptive immune responses and neurogenic inflammation [8]. The ocular surface becomes an immune advanced site once the integrity is dysregulated by a disturbance in the microenvironment on ocular surface. Eye-associated lymphoid tissue (EALT) has been proposed, which is continuous at the ocular surface, including the cornea, conjunctiva, and meibomian gland and its adnexa including the lacrimal glands and lacrimal drainage system [9,10]. A breakdown of EALT homeostasis may result in dry eye disease [11]. This concept is likely to be associated with immune-mediated dry eye disease, including SS, graft-versus-host disease (GVHD) or other conditions of dry eye associated with autoimmune diseases.

Dry eye disease is defined as “a multifactorial disease of the ocular surface characterized by a loss of homeostasis of the tear film, and accompanied by ocular symptoms, in which (1) tear film instability, (2) hyperosmolarity, (3) ocular surface inflammation and damage, and (4) neurosensory abnormalities play etiological roles” by Tear Film Ocular Surface Society Dry Eye Workshop II [12]. On the other hand, the Asia Dry Eye Society proposed a new consensus definition of dry eye disease as “a multifactorial disease characterized by unstable tear film as a core mechanism causing (1) a number of symptoms and/or (2) visual disturbance, potentially accompanied by ocular surface damage” [13]. Recently, International SS diagnostic criteria, The American College of Rheumatology/European League Against Rheumatism Collaborative Initiative (ACR/EULAR) was proposed worldwide [14]. For evaluation of dry eye disease in SS by the criteria, an abnormal ocular staining score of ≥5 (or van Bijsterveld score of ≥4), and the value of Schirmer’s test of ≤5 mm/5 min by ACR/EULAR criteria [14]. It includes the value of Schirmer test and ocular surface staining to evaluate the disease severity and make a diagnosis. On the other hand, Asia Dry Eye Society exclude the staining score of the ocular surface and the value of the Schirmer’s test. Tear Film Ocular Society Dry Eye Workshop II emphasizes the importance of tear osmolarity for core mechanism of dry eye disease. Those issues are different from the diagnosis of ACR/EULAR SS criteria, which require the standardization in diagnosing SS patients internationally in future.

Recent advances in basic and clinical research have increased our understanding of dry eye disease caused by SS [15,16]. Considering the mechanisms underlying SS-related dry eye disease, this review will focus on autoimmune epithelitis of the ocular surface, chronic inflammation and several other molecules in the tear film, cornea, conjunctiva, lacrimal glands, and meibomian glands that represent potential targets in the treatment of SS-related dry eye disease.

## 2. Autoimmune Epithelitis in the Ocular Surface in SS

### 2.1. Autoimmune Epithelitis

Autoimmune epithelitis is explained as epithelial cells of exocrine glands, including the lacrimal glands, possibly meibomian glands, and salivary glands, and mucosal epithelia including conjunctiva are the crucial regulators of the autoimmune response by acting as nonprofessional antigen-presenting cells (APCs) and not just as innocent bystanders as a result of the infiltration of immune cells as one of core mechanistic processes in SS [5,17,18] Aberrant autoantigens are expressed on exocrine gland and mucosal membrane epithelia, which may lead to an increased presentation to autoreactive T cells [17]. Inappropriate events observed in the acinar and ductal epithelial cells indicate the significant role they play and suggest that they may act as antigen presenting cells [18]. Initial intrinsic epithelial activation followed by exocrine gland destruction has been reported to precede lymphocytic infiltration in SS. Salivary gland epithelial cells in SS are reported to play a crucial role in the inflammatory and autoimmune response in humans [6]. There are several lines of evidence insisting the intrinsic cause of autoimmune epithelitis in salivary gland epithelia in SS [6]. In conjunctiva from SS patients, Epstein Barr Virus gene expression, intercellular adhesion molecule (ICAM)-1, human leukocyte antigen (HLA)-DR, and Interleukin (IL)-6 have been found, suggesting that conjunctival epithelia have a functional role in immune response in SS [19]. In addition, HLA-DR and costimulatory molecules are more highly expressed in SS conjunctival epithelia than non SS dry eye disease patients by fluocytometric analysis from impression cytology conjunctival specimens [20]. A previous report showed that HLA-DR and Cluster of Differentiation (CD) 40 expressing acinar epithelial cells in lacrimal gland were attached by CD4^+^ T cells but not destroyed by CD8^+^ T cells in SS patients [21]. The expression of autoantigens is thought to be one of the key triggers of the autoimmune epithelitis in SS [16].

### 2.2. Genetic Predisposition and/or Hormone Factors

Genetic predisposition and/or hormonal factors influence the integrity of glandular epithelia [22]. Recent studies showed that innate and adaptive immune-related genes are upregulated in SS conjunctiva [23] and systemic SS [24]. Hormone imbalance, especially androgen deficiency has been reported to be linked to meibomian gland dysfunction [25] and lacrimal gland dysfunction in SS [25,26]. Effectiveness of treatment by androgen is effective for dry eye disease as has been suggested [26,27,28].

### 2.3. Environmental Factors

As an environmental factor, viral infections, including Epstein-Barr virus (EBV), chronic hepatitis C virus, human immunodeficiency virus, and human T-cell leukemia virus type I, have been implicated in the development of SS epitheliopathy [5]; they can trigger activation in SS epithelia in the lacrimal glands and salivary glands, conjunctiva and possibly the meibomian glands. Immune cells and the inflammatory milieu further activate epithelial cells [5,29,30]. A vicious cycle of epithelial cells and inflammatory cells augment autoimmunity in SS [5,29,30].

### 2.4. Corona Virus Disease-19 (COVID-19) and Autoimmunity

Regarding the recent emergence of the coronavirus disease 2019 (COVID-19), the relationship between SS-related dry eye disease and the virus is largely unknown. On the basis of a review of the literature, Inomata et al. reported that, in samples from COVID-19 patients, 16.7% of conjunctival samples tested positive and tear samples were negative in real-time polymerase chain reaction analysis [31]. In addition, it has been reported that COVID-19 influences immune deficiency and that the presence of rheumatic autoimmune disease is associated with severe COVID-19 [32]. Therefore, close attention should be paid to the association between initiation of SS-related dry eye disease and COVID-19 severity during long-term follow-up in a clinical setting. There is a possibility that epitheliotropic microorganisms play a crucial role for the initiation and chronic activation of epithelia in SS [33].

### 2.5. Ocular Surface and Lacrimal Gland Epithelia and Autoantigen

In a human study, HLA-DR and costimulatory molecules, including CD40 and CD80, necessary for the full component of APCs, have been shown in the lacrimal glands [21], suggesting that SS lacrimal gland epithelia are nonprofessional APCs. However, epithelial cells in several tissues are also known to express major histocompatibility complex (MHC) class II and co-stimulatory molecules in response to inflammatory stimuli that allows them to present their endogenous antigens and epithelial cells are not known to process exogenous antigens like professional APCs with intrinsic ability to present antigens. Therefore, we need to be careful to interpret those results with special attention. Sjögren’s syndrome antigen type A/Ro (Ro 52 and Ro 60), Sjögren’s syndrome antigen type B/La (SSB/La), α-fodrin [34] and β-fodrin as intracellular organ-specific cytoskeleton protein [35], type 3 muscarinic acetylcholine (M3) receptor [36], and the kallikrein family of proteins, including Klk7 [37], and Klk11 [38] in humans and Klk13 [39], and Klk22 [40] in animals, are reported as autoantigens in ocular surface, SS lacrimal glands or salivary glands. Recently, cell-intrinsic activation of ductal epithelia in SS patients was found as persistent epithelial AIM2 (absent in melanoma 2) activation by abnormal cytoplasmic DNA formation [41]. Subsequently, T cell and B cell recruitment, abnormal responses to autoantigens and activation by amplifying cytokine and chemokine production lead to glandular epithelium destruction and severe dry eye disease. In the immune pathogenic process of SS-related dry eye disease, autoimmune epithelitis may be related to immune reaction and antigen presentation of the lacrimal gland epithelia, conjunctival epithelia and possibly the meibomian gland epithelia. For example, the muscarinic type 3 receptor (M3) is found in conjunctiva and meibomian gland epithelia as well as the lacrimal glands [42]. The damaged M3-expressing cells as an autoantigen are associated with a reduction in parasympathetic function [36], which could cause reduced tear fluid or meibum producing function of exocrine glands, leading to dry eye disease in SS [42]. Therefore, M3 receptor as an autoantigen may be linked to autoimmune epithelitis targeted by M3 receptor autoreactive T cells in lacrimal glands, conjunctiva and meibomian glands. The initiation of an adaptive immune response requires that intrinsic autoantigens at the site of inflammation are processed and presented by dendritic cells or macrophages that migrate to regional lymphoid tissue to activate and expand antigen-specific effector T cells [16,43]. As ocular surface tissue from inflammatory conditions is characterized by the upregulation of MHC class II and other costimulatory signals, including CD40, CD80, and CD86, the activation of circulating, primed T cells that are recruited to the cornea and conjunctiva of patients with dry eye disease may occur in SS [39]. Due to altered expression of various inflammatory factors including transforming growth factor-β, IL-6, tumor necrosis factor (TNF)-α, and C-X-C motif chemokine (CXCL) 12 in ocular surface epithelia in a murine model of SS, the authors suggested ocular surface epithelia are targeted by immune reaction [44]. Taken together, the glandular epithelial cells of patients with SS are likely to have the potential to actively participate and regulate the immunopathogenic process in the inflammatory milieu.

### 2.6. Autophagy

Autophagy was found to be a self-eating cellular process to maintain cellular homeostasis via lysosome-mediated degradation in yeast [45,46,47] and reviewed regarding various aspects in immunity [48]. It is also a cell intrinsic recycling system of cytoplasmic components and organelles as a defense mechanism that removes pathogens including virus or bacteria and damaged endocytic compartments from the cytosol [49]. Besides its classic role in response to cellular stress, autophagy has been implicated in the pathogenesis of autoimmune diseases including autoantigen presentation [48]. Autophagy has been found to promote MHC class II presentation of peptides from intracellular source proteins in the human B-lymphoblastoid cell lines in vitro [50]. Bacteria can inhibit lysosomal fusion or maturation by residing in phagosomes, mask themselves or pretend to avoid autophagic recognition. Such behaviors are thought to focus on a basic aspect of autoimmune reaction in differentiating between self and nonself [48].

There is possibility that ocular surface epithelia, including lacrimal gland, conjunctiva, and meibomian gland in SS-related dry eye disease, is associated with autophagy related autoimmune epithelitis. Enhanced autophagy and apoptosis are involved in the SSA(Ro 52 and Ro 60) and SSB/La redistribution in secretory epithelial cells of the salivary gland in patients with SS [51]. Autophagy is reported to regulate various immune processes, such as antigen presentation, pathogen removal, the survival immune cells, and inflammation [48]. During autophagy, a portion of the cytoplasm and several proteins are incorporated in the autophagosome, a key structure with double layer membranes for intracellular degradation. A previous study suggested that autophagy is enhanced or dysregulated in SS by expressing autophagy markers, autophagy related 5 (ATG5) and microtubule-associated protein 1 light chain 3B (LC3II) in tear film, conjunctiva in humans and lacrimal gland in mice. The authors suggested those autophagy makers may serve as both diagnostic and therapeutic biomarkers in SS-related dry eye disease in patients with SS [52]. A dysregulated function of autophagy has recently been reported to be the causation of autoimmune diseases, suggesting a crucial role of autophagy in ocular surface immunology in SS [48].

### 2.7. Translocation of Tear-Secretion-Related Molecules in Ocular Surface Epithelia in SS 

Aquaporins are distributed at the plasma membrane of many cell types throughout organs, playing roles in the body as well as in water and tear secretion (Figure 1). Among the known aquaporin proteins, aquaporin 5 (AQP5) is detected in the cornea and lacrimal glands in ocular lesions as well as the lung and salivary glands in mammalian cells [53]. AQP5 is localized at the apical membrane of acinar cells of lacrimal and salivary glands. AQP5-knockout studies in mice confirmed the pivotal function of AQP5 in water secretion in salivary gland acinar epithelia [54]. Studies have confirmed that abnormal trafficking of AQP5 contributes to the loss of secretory function in vitro in epithelial cells from SS patients [55]. In human studies, healthy controls and Mikulicz’s disease patients as a disease control or non-SS dry eye disease patients had the apical distribution of AQP5 in lacrimal acinar cells [56]. In contrast, diffuse cytoplasmic AQP5 of lacrimal gland epithelia was seen in patients with SS-related dry eye disease patients [56]. These findings show a selective defect of AQP5 distribution in SS lacrimal gland epithelia that might contribute to decreased tear production in SS-related dry eye disease patients.

β-fodrin, a membrane skeleton protein associated with ion channels and pumps, was distributed diffusely in acinar epithelial cell cytoplasm of lacrimal gland in SS-related dry eye patients similar to AQ5 distribution in SS. In contrast, the intact β-fodrin was located at the apical membrane in a disease control, chronic graft-versus-host disease (GVHD)-related dry eye patient. These findings suggest that altered distribution of β-fodrin in glandular epithelial cells of lacrimal gland may reduce secretory function and facilitate an autoimmune response to β-fodrin, leading to glandular damage in SS [35,56]. This translocation may lead to both cause of induction of autoantigen and dysfunctional secretion of tear components.

Vesicle-associated membrane protein 8 (VAMP 8) is a molecule involved in the secretion pathway, and a small GTPase of the ras superfamily (Rab3 D), a hallmark of mature secretory vesicles of epithelia, and is a core regulator of intracellular vesicle transport during exocytosis. Rab3D is reported to be translocated from the apical side to the basal side of lacrimal gland epithelia in SS patients [57], suggesting dysregulated tear secretion and/or dysregulated tear production of SS lacrimal gland epithelia.

The systemic renin–angiotensin system (RAS) plays a key role in the endocrine regulation of blood pressure and salt and water balance. The action of angiotensin II is mediated mainly via its interaction with its two receptor subtypes, angiotensin II type 1 (AT1R) and type 2 (AT2R). Other than systemic RAS, local tissue RAS is recognized in the lacrimal glands in mice [58], as well as the liver, brain, gastrointestinal system, and pancreas. Tissue RAS can be tightly regulated by various physiological and pathological microenvironments. For example, pancreatic RAS is involved in the physiological regulation of pancreatic functions such as exocrine acinar and endocrine islet activities. Tissue RAS and the receptors AT1R and AT2R were found on the basal side and apical side, respectively, suggesting that each receptor regulates the secretion of tear fluid from lacrimal gland epithelia and plays some role in the maintenance of the volume of tear fluid [58]. In immune-mediated diseases, graft-versus-host disease in mice [59], ATR2 is translocated from the apical membrane to the cytoplasm [58,59], similar to aquaporin and β-fodrin trafficking in lacrimal gland epithelia in SS patients.

Based on these studies, altered proteins and other substances probably due to autoimmune epithelitis in SS are likely to be secreted into tear fluid in the SS ocular surface.

Vesicle associated molecular protein 8, Rab3D, aquaporin 5, β-fodrin, and angiotensin II type 2 receptor are translocated from apical membrane to cytoplasma or basal side of lacrimal gland epithelia, resulting in dysregulated production of tear component and loss of tear secretion leads to severe dry eye disease in Sjogren’s syndrome (SS). SS-A and SS-B antigens are also reported to be translocated in SS epithelia.

## 3. Chronic Inflammation of the Ocular Surface and Lacrimal Gland in SS

### 3.1. Tear Film Biomarkers

The tear film consists of various substances, including lipids, proteins, mucins, electrolytes and oxygen [60]. Recent advanced techniques revealed the overexpression of proteins involved in TNF-α signaling (CPNE1), B cell survival (PRDX3) and neutrophil gelatinase-associated lipocalin in tear fluid from SS patients [61]. In addition, ubiquitination (LMO7 and HUWE1) and B cell differentiation (TPD52), the regulation of cellular innate and adaptive immunological pathways, and excessive expression of the proteins including Stomatin (a membrane protein in regulating ion transport; STOM), Annexin A4(ANXA4) and Annexin A1(ANXA1) found in extracellular vesicles are recognized in the tear film of primary SS patients [62]. Autophagy-related molecules, ATG5 andLC3B-II are found to be elevated in the tear film in SS patients [52]. Mucins also lubricate foreign bodies [63]. On the other hand, epidermal fatty-acid binding protein [64], matrix metalloprotease (MMP) 2, MMP9, 8-hydroxy-2′-deoxyguanosine (8-OHdG), and 4-hydroxy-2-nonenal (4-HNE) are elevated in SS tear fluid [63]. On the other hand, decreased levels of lactoferrin, lysosomes, lipocalin, secretory IgA, phospholipase A2, Muc 5Ac, Muc 1, Muc 4, Muc16, and Muc19 have been reported in SS tear fluid [63]. These molecules protect the ocular surface against invading pathogens. As a result, microbes can easily access the ocular surface epithelium. Novel and innovative paper-based microdevice (μPAD) and immunoassay by InflammaDry are useful for measuring lactoferrin concentrations and MMP9, respectively, in human tear fluid affected by dry eye disease including SS [65,66].

### 3.2. Lacrimal Glands

Lacrimal glands as one of the exocrine glands are consisted of a main excretory duct and its branches entering into the hilus of the lobular gland and connecting to the intraglandular structures of the acini. A dense fibrotic interstitium or fibrous capsule surrounded the entire gland, and a variable amount of fibrovascular interstitium demarcated each of the ducts and acini in the gland. Acini and ducts possess their own epithelia containing a variety of proteins that are secreted into ductal lumen.

It is reported that there are more B cells and plasma cells in the lobules of SS-affected lacrimal glands than in those of their chronic GVHD-affected counterparts [67]. In tissue specimens in patients with SS, CD4^+^ and CD8^+^ T cells are equally distributed in acinar areas, with many CD4-CD8-mononuclear cells. CD4^+^ T cells contact closely with SS lacrimal gland epithelia expressing HLA-DR and costimulatory molecules [21], suggesting that SS epithelia present autoantigen to naïve CD4^+^ T cells.

On the other hand, CD8^+^ T cells also have been reported to play an indispensable role for development of pathology through affecting glandular epithelial cells in SS-related dry eye disease patients [68]. A majority of these CD8^+^ T cells are reported to possess a unique integrin, αEβ7 (CD103), and related to apoptotic region of epithelia. This study has been further supported by advanced technology, multiomics analyses [69]. The study identified cytotoxic CD8^+^ T cells as SS gene signatures in whole blood and serum from patients with primary SS and further showed the involvement of cytotoxic CD8^+^ T cells in SS glandular pathology [69]. Both CD4^+^ and CD8^+^ T cells have important roles for the development of SS pathophysiology.

Interferons (IFN)s play an important role for the early innate and later acquired immune stages of SS [15,70]. IFN-γ is expressed in the endothelia of capillaries in SS-disturbed lacrimal gland and conjunctiva, whereas it was expressed on fibroblasts in their GVHD-affected counterparts, suggesting that endothelia of blood vessels may contribute to the development of SS pathogenesis. Several cytokines produced by inflammatory infiltrates, including IFN-γ and IL-17, are amplified, and autoreactive T cells and B cells are activated by IFN in the immune pathogenic process of SS exocrine glands including lacrimal glands [15,37,71,72].

SS often causes lymphoproliferative disorders. Approximately 5% of SS patients develop malignant lymphoma (ML) in long-term follow up. A previous study showed that SS patients were 15–20 times more susceptible to ML than those without SS [2]. The development of lymphoma is reported to be multifactorial and multistep. Chronic stimulation of polyclonal and subsequent activation of B cells may link to an increasing risk of oncogenic mutation and monoclonal selection, and subsequent loss of checkpoints of activation of autoimmune B cells to accelerate the lymphomagenesis [5]. CD30, a member of the TNF receptor superfamily and a hallmark of malignant cells of ML is known to be present in malignant cells of organs affected by ML. Therefore, CD30 expression in lacrimal glands and the conjunctiva in patients with SS may be expressed on B cells and an indicator for the development of ML in SS patients [73]. CD30^+^ cells have already been recognized in ocular tissues in SS patients without ML [73]. CD30 expressing cells are probably B cells which later proliferate as B cell lymphoma. Therefore, great attention should be paid in the long-term follow-up of SS patients. Anti-CD30 antibody is available to treat ML. Anti-CD30 antibody effectiveness in SS should be clarified using an SS animal model, and the function and role of CD30 should be elucidated using CD30-knockout mice.

In clinical practice, IgG4-related ophthalmic disease should be excluded if lacrimal gland swelling is seen [74,75]. Extraglandular manifestations, including neurological signs, skin rashes, interstitial pneumonitis, and lymph node swelling, needs to be examined carefully, in addition to dry eye symptoms. Blood analysis at serum levels of IgG4, anti-SSA/Ro antibody, anti-SSB/La antibody, rheumatoid factor, and anti-ANA antibody is required to discriminate SS- and IgG4-related diseases [74,75].

### 3.3. Conjunctiva

At developmental stage, budding of conjunctival epithelial cells has been shown to be the development of lacrimal gland ducts, lacrimal gland acini and lobules in mice [76,77]. Therefore, some similarities between lacrimal gland epithelia and conjunctival epithelia in terms of molecular expression are present. Conjunctival epithelia in patients with SS express HLA-DR and CD40 [20], full components necessary for antigen presentation similar to lacrimal gland epithelia, suggesting that conjunctival epithelia are also intrinsic APCs, not merely secondary damaged cells due to dryness of ocular surface. The thickness of the membrane spanning mucin is reduced in SS patients compared to normal areas [78]. Microvilli of conjunctival epithelia in SS are distorted and branched [78,79], suggesting reduced production and disoriented guidance of secretory vesicles by microfilaments in secreting membrane spanning mucin in SS patients. Therefore, conjunctival epithelial function may be dysregulated and impaired mucin secretion might be related to dry eye disease in SS. Reduced mucin secretion is reported in SS patients [80,81,82] and mice [83]. Whether impaired mucin secretion causes dry eye disease rather than appearing as an outcome of inflammation in patients SS is subjects of interest. Recently, the important role for immune tolerance in the ocular surface of goblet cells was reported beside the well-known function of gel forming mucin production. Goblet cell loss in dry eye disease is linked to the higher level of Th1 cytokines and IFN-γ expression [84]. It has been reported that goblet cell loss is related to severe dry eye disease including SS as well as ocular cicatricial pemphigoid (OCP), Stevens Johnson syndrome (SJS), and graft-versus-host disease (GVHD), suggesting that goblet cells have immune modulatory functions [84]. On the other hand, there are reports that goblet cell deficient mice develop only milder corneal epithelitis in mice [85]. Further detailed studies are required to understand the goblet cell function in SS-related dry eye disease.

Studies have reported accumulation of potentially long-lived plasma cells in chronically inflamed salivary glands of primary SS patients [86], and SS mouse models [87]. In lacrimal glands of long-term follow up of SS patients, the histopathology shows an excessive accumulation of B cells and plasma cells in conjunctiva and lacrimal gland [21,67,73] which suggested inability of apoptosis, leading to accumulation there. Accumulated plasma cells are likely to produce dysregulated autoantibodies which contribute to the pathological condition and affect the microenvironment of ocular tissue. Subsequent excessive accumulation of B cells and plasma cells suggests that inappropriate apoptosis which is shown as bcl-2^+^ plasma cells in salivary gland in SS patients [86] might be related to the overproduction and deposition of abnormal autoantibodies in the lacrimal gland stromal areas in SS patients [73,86].

Conjunctival epithelia are expressed as acting APCs possibly secondary to viral infection, including EBV and HCV infection [19]. Under electron microscopic observation, microvilli of conjunctival epithelia were branched and shortened in terms of height, suggesting that disoriented microfilaments were directed to microvilli due to dysregulated conjunctival epithelia in patients with SS [78,79]. Dysregulated mucin producibility leads to tear film instability, leading to dry eye disease. CD45^+^HLA-DR APCs, including macrophages, dendritic cells, and B cells, also are associated with the severity of dry eye disease by analyzing impression cytology specimens from patients with SS [88].

In SS-related dry eye disease, changes of quality or decrease of amount in the epithelial glycocalyx which is important for ocular surface barrier [82], a loss of goblet cells, and the keratinization of the conjunctival and corneal epithelia occur combined with the expression of cornified envelope precursor proteins [16]. Squamous metaplasia occurs, where the mucosal membrane is transdifferentiated to an epidermalized surface, and this is observed during the transition to severe dry eye including SS [16]. Small proline-rich proteins in humans and involucrin in human cell lines [89], late envelope proteins and filaggrin are shown as cornified envelope precursor proteins by analyzing impression cytology and cell culture or tissue sections in SS patients [37,90]. It is reported that IL-1β has an essential role in developing squamous metaplasia on the ocular surface epithelia [15] in an animal study [91]. A significant role of IFN-γ is extensively reviewed for human and animal studies regarding mechanism of disease in SS by de Paiva et al. [15]. Both cytokines have been reported to be expressed at the ocular surface in dry eye disease. The expression of the relevant genes has been shown to precede the squamous phenotype. IFN-γ has been reported to promote goblet cell loss, epithelial apoptosis, and the keratinization of the conjunctival epithelium in a dry eye mouse model and play a pivotal role to squamous metaplasia in human dry eye disease [92]. Confocal microscopic analysis suggests a large amount of inflammatory cell infiltration and epithelial structural changes in SS patient conjunctiva [93]. Their further study revealed a link between the expression of reactive oxygen species including lipid peroxidation in conjunctival epithelia and the damage and inflammatory processes in conjunctiva of SS related dry eye patients [94].

SS presented the most severe dry eye disease among other autoimmune diseases with similar severe subjective symptoms and objective findings of cornea and conjunctival dye staining compared to ocular surface of OCP, SJS and ocular GVHD. However, SS patients rarely present cicatrizing conjunctivitis, in contrast to OCP, SJS and ocular GVHD [16]. The difference of fibrotic changes between SS and other three cicatricial ocular diseases may depend on the presence or absence of disturbance in basal lamina of conjunctiva that allow infiltration or migration of pathogenic fibroblasts through disrupted basal lamina as reported in ocular GVHD [16,67].

### 3.4. Cornea

Corneal epithelium is a nonkeratinized and stratified layer with a specific cytokeratin expression pattern in humans [95,96]. Immune response of cornea has been identified, showing that the ocular microenvironment regulates these immune responses by recruiting different populations of inflammatory cells including T cells and macrophages to the cornea [97].

A recent report shows that severe visual impairment was noted in 10%, and 2.5% had corneal complications among the 1838 eyes of primary and secondary SS patients [98]. Corneal scarring (OR 3.00), corneal ulceration (OR 12.96), low Schirmer values are reported [98]. Corneal perforation secondary to corneal ulcer should be avoided, therefore close detailed examination for ocular surface including corneal complications in SS is insisted.

Corneal nociceptors are thought to play an important role as a sentinel for tear evaporation. Corneal neuropathic pain is associated with autoimmune diseases seen predominantly in women, such as SS [99,100]. Corneal nerve abnormalities in the early phase of SS have been reported using in vivo confocal microscopy and may be an initial sign of SS-related dry eye disease [42,101,102]. Therefore, corneal nerves play a pathogenic role of development of severe dry eye in SS. Recently, some discriminating clinical features may help to understand the presence of small-fiber neuropathy in patients with primary SS and chronic neuropathic pain, suggesting that corneal nerve changes may serve as a sign specific for SS [103,104]. Neuroinflammation is associated with dry eye conditions and discomfort in SS. In addition, it is likely that the cold thermoreceptor, Transient Receptor Potential Melastatin 8 (TRPM8), is linked to dry eye disease, particularly to the unpleasantness of dry eye disease related to SS [105]. Neuroinflammation may be related to autoimmune epithelitis because altered antigen presenting cells are present in the ocular surface in patients with SS [15,106]. Neurogenic inflammation may be linked to dry eye disease, especially in SS [8].

### 3.5. Meibomian Glands

Meibomian glands, specialized sebaceous glands, are significantly more damaged in SS patients (57.9%) than in non-SS patients [107]. In addition, the lid margin, located at the margin between the conjunctiva and the skin of the eyelid, is frequently damaged in SS-related dry eye disease patients [107]. Androgen deficiency is a pivotal and causative factor in the pathophysiology of dry eye disease and meibomian gland dysfunction that is related to the development of SS-related dry eye disease [27,28]. Compared with common meibomian gland dysfunction patients, SS patients’ meibomian glands had higher acinar density, smaller diameters, greater density of periglandular inflammatory cells, and lower secretion reflectivity using in vivo confocal microscopy of meibomian glands in SS [108]. The meibomian gland epithelium consists of a mucosal membrane connected to the ocular surface and eye-associated lymphoid tissue. Inflamed meibomian gland dysfunction is reported to confer dry eye disease [109]. Microbiota on the ocular surface can easily access the meibomian gland epithelia in inflamed tissue [29,109]. Meibomian gland dysfunction is reported to exist in other autoimmune diseases including rheumatoid arthritis, systemic sclerosis, and systemic lupus erythematosus [110]. It is likely that autoimmune epithelitis exists in meibomian gland epithelium, similar to conjunctival and lacrimal gland epithelia, although the evidence is still unclear.

### 3.6. Interstitial Stroma: Mesenchymal Cells as Nurse Cells in Immunity

Stromal mesenchymal cells such as fibroblasts are classically thought to produce collagens merely supporting various organs, but a body of evidence supports that those mesenchymal cells play some role to act as immunosurveillance or immune cells themselves as antigen presenting cells. For example, fibrocytes are reported to appear several times more potent than dendritic cells for antigen presentation [111].

CD34^+^ fibroblasts are reported to reside in mammary glands, submandibular glands and thyroid probably acting in immune surveillance [112,113,114], and lacrimal gland and conjunctiva in immune-mediated dry eye disease [67,115,116]. Those fibroblasts or mesenchymal cells in the stroma may play some role as maintaining the homeostasis of ocular surface including lacrimal gland microenvironment in healthy condition, and act as regenerative or pathogenic trigger of inflamed tissue [117]. In immune-mediated dry eye disease, including systemic autoimmune diseases such as SS, rheumatoid arthritis and systemic sclerosis and GVHD, those fibroblasts may be partially derived from bone marrow and activated to trigger immune-mediated fibrosis in autoimmune disease by interacting with immune competent cells including macrophages, T cells, or B cells in patients with chronic GVHD with similar findings of dry eye disease in SS [21,116] and in sclerodermatous mouse models [115]. Although SS-related fibrosis progresses gradually, bone-marrow-derived stromal cells might trigger and support T cell and B cell activation and maturation [118] because these stromal/stem cells interact with immune competent cells at various sites of the ocular surface. It may be useful to revisit the stromal mesenchymal cells, whether they are linked to pathogenetic properties in SS-related dry eye disease.

## 4. Various Components in Causing Sjogren’s Syndrome Related Dry Eye Disease

Based on the previous reports and findings, we propose the following model of pathogenic process in SS-related dry eye disease. Possible pathophysiology of SS-related dry eye disease may be initiated by autoimmune epithelitis in lacrimal gland, conjunctiva and meibomian gland epithelia under genetic predisposition, environmental factors and hormone imbalance. Viral infection is one of the candidates to initiate the abnormal expression of various proteins as SSA, SSB, M3 receptor, α-fodrin and Kallikrein. HLA-DR, adhesion molecules, and co-stimulatory molecules in conjunctival and lacrimal gland epithelia. Dysregulated autophagy plays some role in presenting autoantigens and redistribution and translocation of proteins related to tear component secretion molecules including β-fodrin, AQP5, Rab3D, and Angiotensin II type I receptor in lacrimal gland epithelia. CD4^+^ T cells are observed to attach to the basal side of epithelia, suggesting that lacrimal gland and ocular surface epithelia present autoantigens to CD4^+^ T cells as nonprofessional antigen presenting cells. On the other hand, it is suggested that CD8^+^ T cells also play some role to destroy the epithelia through the perforin-granzyme pathway. In long term SS patients, accumulation of B cells and plasma cells in the lacrimal gland stroma by producing excessive autoantibodies and accumulating abnormal autoantigens are observed (Figure 2).

The pathophysiology of Sjogren’s syndrome related dry eye disease may be initiated by autoimmune epithelitis in the lacrimal gland, conjunctiva and possibly meibomian gland epithelia. Viral infection including Epstein Barr Virus (EBV), Hepatitis C virus (HCV), or Human T cell leukemia virus (HTLV)-1 are possible candidates to initiate the abnormal expression of various autoantigens including Sjogren’s syndrome antigen type A (SSA/Ro), Sjogren’s syndrome antigen type B (SSB/La), M3 muscarinic acetylcholine receptor (M3R), or α-fodrin with HLA-DR, adhesion molecules, and co-stimulatory molecules in the ocular surface and lacrimal gland epithelia. Translocation of various proteins related to tear component secretion may be influenced by dysregulated autophagy in SS. On the other hand, it is suggested that CD8^+^ T cells also play an important role to destroy the epithelia. In long-term follow-up of SS patients, there is accumulation of plasma cells in the stroma, leading to ocular surface epithelium dysfunction, resulting in severe dry eye disease. Proposed by Sumida T, et al. with some modification [119].

## 5. Future Direction

Although SS is classified as “aqueous tear-deficient dry eye”, damage of multiple components, including aqueous tear deficiency, mucin deficiency, and tear evaporation, especially in severe dry eye patients suffering from SS, are frequently observed. In addition, careful attention is required for the possible progression to malignant lymphoma in lid swelling in SS patients. The accumulation of evidence has substantially progressed, but further investigation will be required to clarify the molecular mechanism of the development of SS-related dry eye disease and to overcome this intractable disease. There may be various potential targets which are related to autoimmune epithelitis of the ocular lesion, chronic inflammation, and several other molecules in the tear film, cornea, conjunctiva, lacrimal glands, and meibomian glands in early diagnosis, prevention and treatment of SS-related dry eye disease. More comprehensive understanding of the dry eye disease process and understanding the pathophysiological signaling pathways in SS-related dry eye disease will help develop more innovative, effective and targeted therapies for these patients in future.

## Figures and Tables

**Figure 1 ijms-22-11820-f001:**
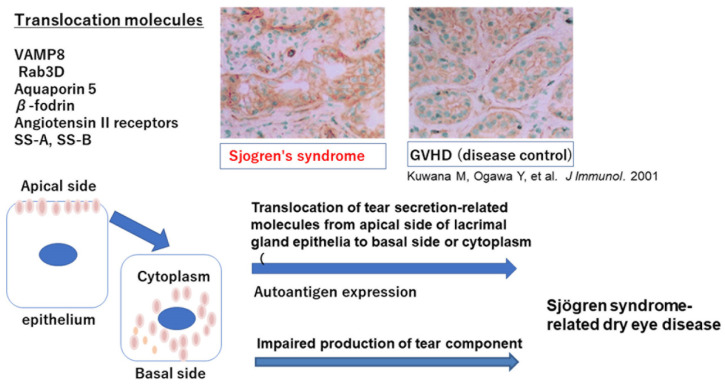
Translocation of tear secretion-related molecules from apical membrane to cytoplasma or basal side.

**Figure 2 ijms-22-11820-f002:**
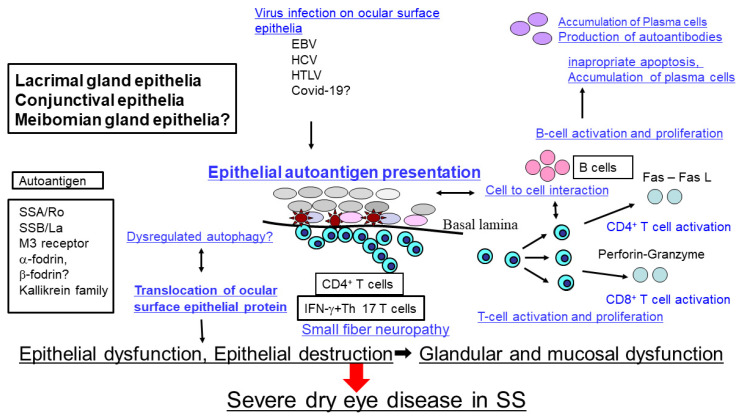
Hypothetical schema of pathophysiology of Sjogren’s syndrome related dry eye disease.

## Data Availability

The data that support the findings of this study are available within the article or from the corresponding author upon reasonable request.

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
