# Peer review of "Autoimmune Epithelitis and Chronic Inflammation in Sjögren’s Syndrome-Related Dry Eye Disease"

_ijms, 2021, doi:10.3390/ijms222111820_

Round 1
Reviewer 1 Report
Detailed review of current understanding of various etiologies of Sjogren's syndrome and other autoimmune diseases causing dry eye disease. The degree of detail and referencing is excellent. However the paper does not pull these various mechanisms together sufficiently. The reader is left with a list of various causes of dysfunction of specific parts of the eye ( epithelial cells, meibomian glands etc. without a discussion pulling this all together.1) Please check and add complete wording of abbreviations including MHC, HLA- DR , TNF etc at first use. 2) Adjust font page 3. 3) There should be a section discussing and postulating how the various causes of the components of dry eye disease listed in section 3 fit together in causing Sjogrens specifically. This can be section 4. 4) There should be a longer conclusion labeled section 5. to draw the paper to a close.
Author Response
Re: Resubmission of IJMS; Manuscript ID: ijms-1113031; Title: Autoimmune epithelitis and chronic inflammation in Sjögren’s syndrome-related dry eye disease: Authors: Yoko Ogawa *, Tsutomu Takeuchi, Kazuo Tsubota”; Special issue: Submitted to section: Molecular Pathology, Diagnostics, and Therapeutics,
Dry Eye and Ocular Surface Disorders 3.0
Response to Reviewer 1 Comment
Comments and Suggestions for Authors
Detailed review of current understanding of various etiologies of Sjogren's syndrome and other autoimmune diseases causing dry eye disease. The degree of detail and referencing is excellent.
However the paper does not pull these various mechanisms together sufficiently.
The reader is left with a list of various causes of dysfunction of specific parts of the eye ( epithelial cells, meibomian glands etc. without a discussion pulling this all together.
- Please check and add complete wording of abbreviations including MHC, HLA- DR , TNF etc at first use.
We spelled out the abbreviations at first use in the manuscript as follows.
Abstract: The phrase, “ MHC class II or HLA-DR” is replaced with “major histocompatibility complex class II or human leukocyte antigen-DR”.
Line 12 MHC class II was replaced with “major histocompatibility antigen” class II.
Line 12 HLA-DR was spelled out as “human leukocyte antigen (HLA”)-DR at first use in the manuscript.
Line 85. intercellular adhesion molecule (ICAM)-1, human leukocyte antigen (HLA)-DR,
Lines 115-116. major histocompatibility complex (MHC) class II
Line 145 “tumor necrosis factor -“ is replaced as “tumor necrosis factor (TNF)”.
- Adjust font page 3.
Thank you for pointing out this point. We have revised the font for the second paragraph in page 3.
- There should be a section discussing and postulating how the various causes of the components of dry eye disease listed in section 3 fit together in causing Sjogrens specifically. This can be section 4.
We appreciate the reviewer’s comments. We have revised our manuscript with a new section 4 titled as “Various components in causing Sjogrens’s syndrome related dry eye disease.”as suggested the reviewer 1 as follows.
Lines 444-482
“4 Various components in causing Sjogrens’s syndrome related dry eye disease.
Based on the previous reports and findings, we propose the following model of pathogenic process in Sjogren’s syndrome (SS)-related dry eye disease. Possible pathophysiology of SS related dry eye disease may be initiated by autoimmune epithelitis in lacrimal gland, conjunctiva and meibomian gland epithelia under genetic predisposition and hormone imbalance. Viral infection is one of the candidates to initiate the abnormal expression of various proteins as SSA, SSB, M3 receptor, a-fodrin and Kallikrein. HLA-DR, adhesion molecules, and co-stimulatory molecules are expressed in conjunctival and lacrimal gland epithelia. Dysregulated autophagy play some role in presenting autoantigen and redistribution and translocation of protein related to tear component secretion molecules including b-fodrin, AQP5, Rab3D, and Angiotensin II type I receptor in lacrimal gland epithelia. CD4+ T cells are observed by attaching basal side of epithelia, suggesting that lacrimal gland and ocular surface epithelia present autoantigens to CD4+ T cells as non professional antigen presenting cells. On the other hand, it is suggested that CD8+ T cells also play some role to destroy the epithelia through perforin-granzyme pathway. In long term SS patients, accumulation of B cells and plasma cells in the stroma by producing excessive autoantibodies and accumulating abnormal autoantigens. (Figure 2)
Figure 2. Hypothetical shcema of pathophysiology of Sjogren’s syndrome related dry eye disease.
The one of the pathophysiology of Sjogren’s syndrome related dry eye disease may be initiated by autoimmune epithelitis in lacrimal gland, conjunctiva and possibly meibomian gland epithelia. Viral infection including Epstein Barr Virus (EBV), Hepatitis C virus (HCV), or Human T cell leukemia virus (HTLV) are possible candidates to initiate the abnormal expression of various autoantigen including Sjogren’s syndrome antigen type A (SSA), Sjogren’s syndrome antigen type B (SSB) , Muscarinic receptor type 3 (M3), or a-fodrin with HLA-DR, adhesion molecules, and co-stimulatory molecules in ocular surface and lacrimal gland epithelia.Translocation of various proteins related to tear component secretion may be influenced by dysregulated autophagy in SS. On the other hand, it is suggested that CD8+ T cells also play an important role to destroy the epithelia. In long term SS patients, accumulation of plasma cells in the stroma, leading to ocular supface epithelial dysfunction, resulting in severe dry eye disease. Proposed by Sumida T, et al with some modification.
4) There should be a longer conclusion labeled section 5. to draw the paper to a close.
We have revised our manuscript with a longer conclusion labeled section 5 as suggested the reviewer 1.
Lines 514-527
“5. Future direction
Although SS is classified as “aqueous tear-deficient dry eye”, multiple components, including aqueous tear deficiency, mucin deficiency, and tear evaporation, especially in severe dry eye patients suffering from SS, are frequently damaged. In addition, careful attention is required for the possible progression to malignant lymphoma in lid swelling in SS patients. The accumulation of evidence has substantially progressed, but further investigation will be required to clarify the molecular mechanism of the development of SS-related dry eye disease and to overcome this intractable disease. There may be various potential targets which is related to autoimmune epithelitis of the ocular lesion, chronic inflammation, and several other molecules in the tear film, cornea, conjunctiva, lacrimal glands, and meibomian glands in early diagnosing, preventing and treating SS-related dry eye disease. More comprehensive understanding of dry eye disease process and understanding the pathophysiological signaling pathways in SS related dry eye disease will develop more innovative, effective and targeted therapies for these patients in future.”
Submission Date
18 February 2021
Date of this review
19 Feb 2021 20:50:29

Reviewer 2 Report
This review aims to focus on the role of autoimmune epithelitis in dry eye disease associated with Sjögren’s syndrome in the context of chronic inflammation and review of potential therapeutic targets. However, manuscript fails to focus on highlighted points in the abstract and present conceptually organized information. Several ideas are difficult to follow due to grammatical errors.
Below are some major concerns -
- Epithelitis in SS pathology is extensively reviewed in the context of Salivary gland epithelial cells. As such the first paragraph under second sub-heading cites references regarding salivary gland epithelial cells to claim their role in autoimmune pathology and assumes its extension to lacrimal glands. In order to describe similar dysregulation of epithelial cells in ocular surface disease it will be helpful to define autoimmune epithelitis clearly and identify ocular surface specific literature.
- Although TFOS and Asia Dry Eye Society definitions of dry eye disease are mentioned there is no attempt made to explain the distinctions between dry eye associated with Sjögren’s disease vs. that associated with other diseases.
- It is unclear if authors are suggesting ocular surface epithelitis in Sjögren’s syndrome as a cause or an effect of the pathology. Also, justification for including lacrimal gland along with ocular surface tissues is not provided adequately.
- Description of lacrimal gland epithelial cells as “intrinsic APCs” on page 2 under second sub-heading is incorrect and inconsistent with the description as “nonprofessional APCs” included in the abstract. Epithelial cells in several tissues are known to express MHC class II and co-stimulatory molecules in response to inflammatory stimuli that allows them to present their endogenous antigens. But they are not known to process exogenous antigens like professional APCs with intrinsic ability to present antigens. Thus basic description of epithelial cells and their function under normal and pathological conditions needs to be presented clearly and accurately.
- On page 2 (first sentence) “M3 receptor is found in conjunctiva and meibomian gland epithelia…..” - it is unclear what example does this statement represent and how it is related to prior statements.
- Several statements in the review are vague and do not include appropriate citations. For example; on page 2 – “Immune cells and inflammatory milieu further activate epithelial cells or associate with their survival.”, on page 3 (last sentence) – “A dysregulated function of autophagy has been linked to causation and prevention of autoimmune diseases” is a contradictory sentence and does not “suggest a crucial role of autophagy in ocular surface immunology in SS” as stated,
- On page 3 – The statement about abnormal trafficking of AQP5 and anti-M3R antibody antagonism does not make sense. Also the link or similarity between beta-fodrin and AQP5 is unclear. It will be helpful to clarify whether authors consider translocations of proteins to introduce autoantigens or to cause specific functional loss.
- On page 5 – There is no reference cited to support statement regarding decreased levels of lactoferrin, lysosomes, lipocalin ……etc.
- On page 5, sub-heading 3.2 – The description of adaptive immune response is not adequately linked to epithelitis. References to B cells and CD30 under this sub-heading appears abrupt and unclear. What cells in the lacrimal glands express CD30?
- On page 6, sub-heading 3.3
- it is not clear what is meant by “internalization” of conjunctival epithelial cells.
- no citation is provided for MHC class II and CD86 expression by conjunctival epithelial cells and their ability to present antigen.
- no citation is provided to support conclusion that impaired mucin secretion causes dry eye disease rather than appearing as an outcome of inflammation. In fact goblet cell deficient mice were reported to develop only milder corneal epitheliopathy.
- it is not clear how excessive accumulation of B cells and plasma cells is suggestive of inappropriate apoptosis that leads to overproduction of and deposition of abnormal autoantibodies.
- are conjunctival abnormalities different in SS, ocular cicatricial pemphigoid, Steven-Johnson syndrome and GVHD? Needs further clarification of similarities or differences.
- the last two statements in this sub-heading are unclear and need better integration if they are related to the topic covered in the subheading.
- On page 7, sub-heading 3.4
- statement about visual impairment and corneal complications among 1838 eyes makes no sense. What eyes are being referred to here?
- how does demonstration of the relevance of corneal nerve ablation in immune privilege support their “pathogenic” role in SS dry eye.
- On page 8, sub-heading 3.6
- what is “immune-mediated dry eye disease”? is this a different subset of dry eye disease?
- reference to “figure candidate” is unclear.
- Future directions are vague and refer to a generic need to clarify molecular mechanisms of SS-related dry eye disease development. Authors need to provide their specific views here with regards to autoimmune epithelitis that they intended to focus on but failed to do so effectively.
- Overall, throughout the manuscript it is important to state clearly information known from experimental animal models and whether it has been corroborated in human SS patients. Stating observations from animal studies as conclusive findings in SS pathology is confusing and misleading.
Author Response
Re: Resubmission of IJMS; Manuscript ID: ijms-1113031; Title: Autoimmune epithelitis and chronic inflammation in Sjögren’s syndrome-related dry eye disease: Authors: Yoko Ogawa *, Tsutomu Takeuchi, Kazuo Tsubota”; Special issue: Submitted to section: Molecular Pathology, Diagnostics, and Therapeutics,
Dry Eye and Ocular Surface Disorders 3.0
Response to Reviewer 2 Comment
Comments and Suggestions for Authors
This review aims to focus on the role of autoimmune epithelitis in dry eye disease associated with Sjögren’s syndrome in the context of chronic inflammation and review of potential therapeutic targets. However, manuscript fails to focus on highlighted points in the abstract and present conceptually organized information. Several ideas are difficult to follow due to grammatical errors.
Below are some major concerns –
We would like to express our gratitude to all Editors and Reviewers for taking their time and providing us important feedback. We updated our paper according to the reviewer’s comments and prepared by point by point response to all requests. The reviewer’s comments are written in bold and italic. Our response is written in common style.
- Epithelitis in SS pathology is extensively reviewed in the context of Salivary gland epithelial cells. As such the first paragraph under second sub-heading cites references regarding salivary gland epithelial cells to claim their role in autoimmune pathology and assumes its extension to lacrimal glands. In order to describe similar dysregulation of epithelial cells in ocular surface disease it will be helpful to define autoimmune epithelitis clearly and identify ocular surface specific literature.
We explained autoimmune epithelitis as follows.
Lines 69-77 (Line number indicates the exact line number of clear version of the revised manuscript.)
Autoimmune epithelitis is explained asepithelial cells of exocrine glands, including the lacrimal glands, possibly meibomian glands, and salivary glands, and mucosal epithelia including conjunctiva are the crucial regulators of the autoimmune response by acting as nonprofessional antigen-presenting cells (APCs) and not just as innocent bystanders as a result of the infiltration of immune cells as one of core mechanistic process in SS 5, 17, 18 Aberrant autoantigens are expressed on exocrine gland and mucosal membrane epithelia, which may lead to an increased presentation to autoreactive T cells 17. Innappropriate events observed in the acinar and ductal epithelial cells which indicate a significant role they play and which suggest that they may act as antigen presenting cells 18.
As for ocular surface specific literature, we described as follows with citations.
Lines 83-91
In conujunctiva from SS patients, Epstein Barr Virus gene expression, Intercellular adhesion molecule (ICAM)-1, Human Leukocyte Antigen (HLA)-DR, and Interleukin(IL)-6 has been found, suggesting that conjunctival epithelia have a functional role in immune response in SS 19.In addition, HLA-DR and costimulatory molecules are more highly expressed in SS conjunctival epithelia than non SS dry eye disease patients. by fluocytometric analysis form impression cytology conjunctival specimens 20. HLA-DR and Cluster of Differentiation (CD) 40 expessing acinar epithelial cells in lacrimal gland are attached by CD4+ T cells but not destroyed by CD8+ T cells were shown in SS patients21 .
- Although TFOS and Asia Dry Eye Society definitions of dry eye disease are mentioned there is no attempt made to explain the distinctions between dry eye associated with Sjögren’s disease vs. that associated with other diseases.
We deeply appreciate the reviewers pointed out this issue. We revised this portion as follows.
Lines 43-61
“Dry eye disease is defined as “a multifactorial disease of the ocular surface characterized by a loss of homeostasis of the tear film, and accompanied by ocular symptoms, in which 1) tear film instability, 2)hyperosmolarity, 3) ocular surface inflammation and damage, and 4) neurosensory abnormalities play etiological roles” by Tear Film Ocular Surface Society Dry Eye Workshop II 12. On the other hand, the Asia Dry Eye Society proposed a new consensus definition of dry eye disease as “a multifactorial disease characterized by unstable tear film as a core mechanism causing 1) a plenty of symptoms and/or 2) visual disturbance, potentially accompanied by ocular surface damage” 13.Recently, International SS diagnostic criteria, The American College of Rheumatology/European League Against Rheumatism Collaborative Initiative (ACR/EULAR) was proposed worldwide 14. For evaluation of dry eye disease in SS by the criteria, an abnormal ocular staining score of ≥5 (or van Bijsterveld score of ≥4), and a Schirmer's test result of ≤5 mm/5 minutes by ACR/EULAR criteria 14. It includes the value of Schirmer test and ocular surface staining to evaluate the disease severity and make a diagnosis. On the other hand, Asia Dry Eye Society exclude the staining score of ocular surface and the value of the Schirmer’s test. Tear Film Ocular Society Dry Eye Workshop II insists the importance of tear osmolarity for core mechanism of dry eye disease. Those issues are different from the diagnosis of ACR/EULAR SS criteria, which require the standardization in diagnosing SS patients internationally in future.”
- It is unclear if authors are suggesting ocular surface epithelitis in Sjögren’s syndrome as a cause or an effect of the pathology. Also, justification for including lacrimal gland along with ocular surface tissues is not provided adequately.
We appreciate the crucial comments. It is still determined whether ocular surface epithlitis in SS as a cause or an effect of pathology, however, there are several reports insisting that intrinsic cause of autoimmune epithelitis in salivary gland epithelia in SS. In addition, HLA-DR and costimulatory molecules are more highly expressed in SS conjunctival epithelia and lacrimal glands compared to non SS dry eye disease. We have incorporated the reviewer’s precious comments into the sentence as follows.
Lines 81-92
“It is still determined whether ocular surface epithelitis in SS as a cause or an effect of pathology, there are several evidence insisting that intrinsic cause of autoimmune epithelitis in salivary gland epithelia in SS 6. In conujunctiva from SS patients, Epstein Barr Virus gene expression, Intercellular adhesion molecule (ICAM)-1, Human Leukocyte Antigen (HLA)-DR, and Interleukin(IL)-6 has been found, suggesting that conjunctival epithelia have a functional role in immune response in SS 19.In addition, HLA-DR and costimulatory molecules are more highly expressed in SS conjunctival epithelia than non SS dry eye disease patients. by fluocytometric analysis form impression cytology conjunctival specimens 20. HLA-DR and Cluster of Differentiation (CD) 40 expessing acinar epithelial cells in lacrimal gland are attached by CD4+ T cells but not destroyed by CD8+ T cells were shown in SS patients21”
We added a sentence and reference regarding lacrimal gland epithelisis as follows.
Lines 90-92
“HLA-DR and Cluster of Differentiation (CD) 40 expessing acinar epithelial cells in lacrimal gland are attached by CD4+ T cells but not destroyed by CD8+ T cells were shown in SS patients21”
- Description of lacrimal gland epithelial cells as “intrinsic APCs” on page 2 under second sub-heading is incorrect and inconsistent with the description as “nonprofessional APCs” included in the abstract. Epithelial cells in several tissues are known to express MHC class II and co-stimulatory molecules in response to inflammatory stimuli that allows them to present their endogenous antigens. But they are not known to process exogenous antigens like professional APCs with intrinsic ability to present antigens. Thus basic description of epithelial cells and their function under normal and pathological conditions needs to be presented clearly and accurately.
We deeply appreciate the precious instructions. We incorporated the reviewer’s logical statements in our manuscript as follows.
Lines 112-119
“In human study, HLA-DR and costimulatory molecules, including CD40 and CD80, necessary for the full component of APCs, have been shown in the lacrimal glands 21, suggesting that SS lacrimal gland epithelia are non professional APCs. However, epithelial cells in several tissues are also known to express MHC class II and co-stimulatory molecules in response to inflammatory stimuli that allows them to present their endogenous antigens and epithelial cells are not known to process exogenous antigens like professional APCs with intrinsic ability to present antigens. Therefore, we need to be careful to interpret those results with special attention.”
- On page 2 (first sentence) “M3 receptor is found in conjunctiva and meibomian gland epithelia…..” - it is unclear what example does this statement represent and how it is related to prior statements.
I apologize my insufficient explanation. We stated the M3 receptor is potent as autoantigen. We added the sentences as follows by adding reference.
Lines 131-137
“Immune-mediated destruction of the M3-expressing cells as an autoantigen has been reported to be associated with a reduction in parasympathetic function, which could cause reduced tear fluid or meibum producing function of exocrine glands including lacrimal glands and meibomian glands, leading to dry eye disease in SS. 37. Therefore, M3 receptor as an autoantigen may be linked to autoimmune epithelitis targeted by M3 receptor autoreactive T cells in lacrimal glands, conjunctiva and meibomian glands.”
- Several statements in the review are vague and do not include appropriate citations.
For example; on page 2 – “Immune cells and inflammatory milieu further activate epithelial cells or associate with their survival.”
We deleted the following phrase form the sentence “or associate with their survival”.
Line 99-100. “Immune cells and the inflammatory milieu further activate epithelial cells or associated with their survival 5 23 24.
We added an important reference regarding on inflammation and dry eye disease as follows.
No. 23 Pflugfelder SC, de Paiva CS. The Pathophysiology of Dry Eye Disease: What We Know and Future Directions for Research. Ophthalmology. 2017;124(11s):S4-s13.
No. 24. Yamaguchi T. Inflammatory Response in Dry Eye. Invest Ophthalmol Vis Sci. 2018;59(14):Des192-des199.
on page 3 (last sentence) – “A dysregulated function of autophagy has been linked to causation and prevention of autoimmune diseases” is a contradictory sentence and does not “suggest a crucial role of autophagy in ocular surface immunology in SS” as stated,
I apologize my mistake for description of the contradictory sentence. I deleted a phrase “ and prevention “ and cited a reference.
Lines 173-174
“A dysregulated function of autophagy has recently been reported to be the causation and prevention of autoimmune diseases, suggesting a crucial role of autophagy in ocular surface immunology in SS43. ”
No. 43. Zhou XJ, Zhang H. Autophagy in immunity: implications in etiology of autoimmune/autoinflammatory diseases. Autophagy. 2012;8(9):1286-1299.
- On page 3 – The statement about abnormal trafficking of AQP5 and anti-M3R antibody antagonism does not make sense.
We deleted the sentence regarding anti-M3R antibody.
Also the link or similarity between beta-fodrin and AQP5 is unclear. It will be helpful to clarify whether authors consider translocations of proteins to introduce autoantigens or to cause specific functional loss.
We consider the dislocation of both beta-fodrin and AQP5 is thought to be introduce autoantigens and then cause specific functional loss of tear component secretion.
We revisedthe following sentences to explain beta fodin and aquaporin 5 translocation as follows.
Lines 186-199
“In human study, healthy controls and Mikulicz's disease patients as a disease control or non-SS dry eye disease patients had the apical distribution of AQP5 in lacrimal acinar cells. In contrast, diffuse cytoplasmic AQP5 of lacrimal gland epithelia was seen in patients with SS related dry eye disease patients. These findings show a selective defect in epithelia of lacrimal gland AQP5 distribution in SS that might contribute to decreased tear production in dry eye disease in SS patients.
ꞵ-fodrin, a membrane skeleton protein associated with ion channels and pumps, was distributed diffusely in acinar epithelial cell cytoplasm of lacrimal gland in SS related dry eye patients similar to AQ5 distribution in SS. In contrast, the intact ꞵ-fodrin was located at apical membrane in a disease control, chronic graft-vs-host disease related dry eye patients These findings suggest that altered distribution of ꞵ-fodrin in glandular epithelial cells of lacrimal gland may induce reduced secretory function and facilitate an autoimmune response to ꞵ-fodrin, leading to glandular damage in SS.30, 51This translocation may lead to both cause of induction of autoantigen and dysfunctional secretion of tear components.”
We incorporated the explanation according to the reviewer’s suggestion.
Lines 198-199.
“Those translocation may leads to cause of both induction of autoantigen and dysfunctional secretion of tear components.”
- On page 5 – There is no reference cited to support statement regarding decreased levels of lactoferrin, lysosomes, lipocalin ……etc.
Thank you for pointing out this issue. We cited the reference here.
“On the other hand, decreased levels of lactoferrin, lysosomes, lipocalin, secretory IgA, phospholipase A2, Muc 5Ac, Muc 1, Muc 4, Muc16, and Muc19 have been reported in SS tear fluid 58”
No. 58. Narayanan S, Redfern RL, Miller WL, et al. Dry eye disease and microbial keratitis: is there a connection? The ocular surface. 2013;11(2):75-92.
- On page 5, sub-heading 3.2 – The description of adaptive immune response is not adequately linked to epithelitis. References to B cells and CD30 under this sub-heading appears abrupt and unclear. What cells in the lacrimal glands express CD30?
This part, Sub-heading 3.2 is related to chronic inflammation following autoimmune epithelitis. We revised subheading 3 as “Chronic inflammation of the ocular surface and lacrimal gland in SS”.
I tried to describe the later stage of chronic inflammation following autoimmune epithelitis in SS in this section. This reference is important because CD30 positive cells, a marker of malignant lymphoma, have already recognized in SS patient without malignant lymphoma. CD30 expressing cells are probably B cells which later proliferate as B cell lymphoma.
Lines 288-297
“CD30, a member of the tumor necrosis factor receptor superfamily and a hallmark of malignant cells of Hodgkin lymphoma known as Reed-Sternberg cells, is known to be present in malignant cells of organs affected by Hodgkin’s lymphoma or to serve as a prognostic marker of diffuse large B cell lymphoma. Therefore, CD30 expression in lacrimal glands and the conjunctiva in patients with SS may be expressed on B cells and an indicator for the development of ML in SS patients 68. CD30+ cells have already recognized in ocular tissues in SS patient without malignant lymphoma. CD30 expressing cells are probably B cells which later proliferate as B cell lymphoma. Therefore, great attention should be paid in the long-term follow-up SS patients.”
- On page 6, sub-heading 3.3
- it is not clear what is meant by “internalization” of conjunctival epithelial cells.
I apologize the word is unclear. I rewrote the sentence as follows.
Lines 308-309
At developmental stage, budding of conjunctival epithelial cells has been shown to be the development of lacrimal gland ducts, lacrimal gland acini and lobules in mice71 72.
- no citation is provided for MHC class II and CD86 expression by conjunctival epithelial cells and their ability to present antigen.
I am sorry for citation is not provided here. Those data are our unpublished data.
Line 311-312. “Conjunctival epithelia in patients with SS express HLA-DR and CD86(Ogawa Y, et al. unpublished paper),”
Lines 87-89
“HLA-DR and costimulatory molecules are more highly expressed in SS conjunctival epithelia than non SS dry eye disease patients. by fluocytometric analysis form impression cytology conjunctival specimens 20.”
- no citation is provided to support conclusion that impaired mucin secretion causes dry eye disease rather than appearing as an outcome of inflammation. In fact goblet cell deficient mice were reported to develop only milder corneal epitheliopathy.
Lines 320-328
“Although reduced mucin secretion is reported in SS patients75, 76 77 and mice 78, whether impaired mucin secretion causes dry eye disease rather than appearing as an outcome of inflammation in patients SS is not elusive. It has been reported that goblet cell loss is related to severe dry eye disease including SS as well as OCP, SJS, and GVHD, suggesting that goblet cell have immune modulatory functions 79. On the other hand, there are reports that goblet cell deficient mice were reported to develop only milder corneal epithelits in mice 80. Further detailed studies are required to understand the goblet cell function in SS related dry eye disease.”
- it is not clear how excessive accumulation of B cells and plasma cells is suggestive of inappropriate apoptosis that leads to overproduction of and deposition of abnormal autoantibodies.
We appreciate the reviewers important suggestions
It has been reported that accumulation of potentially long-lived plasma cells in chronically inflamed salivary glands of primary SS patientsand SS mouse model. In lacrimal gland of long-term SS patient, the histopathology shows an excessive accumulation of B cell and plasma cells which suggested inability of apoptosis and accumulating there. Accumulated plasma cells are likely to produce dysregulated autoantibodies which contribute to pathological condition, and affect the microenvironment of ocular tissue.
We modified the following portion as follows.
Lines 329-338.
It has been reported that accumulation of potentially long-lived plasma cells in chronically inflamed salivary glands of primary SS patients 81and SS mouse model. 82 In lacrimal gland of long-term SS patient, the histopathology shows an excessive accumulation of B cell and plasma cells in conjunctival and lacrimal gland 21, 62, 68which suggested inability of apoptosis and accumulating there. Accumulated plasma cells are likely to produce dysregulated autoantibodies which contribute to pathological condition, and affect the microenvironment of ocular tissue. Subsequent excessive accumulation of B cells and plasma cells suggests that inappropriate apoptosis which is shown as bcl-2+ plasma cells in salivary gland in SS patients 81 might related to the overproduction and deposition of abnormal autoantibodies in the areas in SS patients68 81.
”
- are conjunctival abnormalities different in SS, ocular cicatricial pemphigoid, Steven-Johnson syndrome and GVHD? Needs further clarification of similarities or differences.
SS presented most severe dry eye disease among other autoimmune disease with similar several severe symptoms and cornea and conjunctival dye staining to OCP, SJS and GVHD. However, SS patients rarely present cicatrizing conjucntivitis, in contrast to ocular cicatricial pemphigoid, Stevens Johnson syndrome and GVHD. We incorporated those description in the revised manuscript as follows.
Lines 368-376
“SS presented most severe dry eye disease among other autoimmune disease with similar severe subjective symptoms and objective findings of cornea an conjunctival dye staining to ocular cicatricial pemphigoid (OCP), Stevens Johnson syndrome (SJS) and ocular graft-versus-host disease (GVHD). However, SS patients rarely present cicatrizing conjucntivitis, in contrast to OCP, SJS and ocular GVHD15. The difference of fibrotic changes between SS and other three cicatricial ocular diseases may depend on the presence or absence of disturbance in basal lamina of conjunctiva that allow infiltration or migration of pathogenic fibroblasts through disrupted basal lamina as reported in ocular GVHD 15 57.”
- the last two statements in this sub-heading are unclear and need better integration if they are related to the topic covered in the subheading.
I apologize those unclear sentences. I corrected the two sentences as follows. The phrases in red characters are modified portions. Lines 363-367
“Confocal microscopic analysis suggest that a large number of inflammatory cell infiltration, and epithelial structual changes in SS conjunctiva 78 Thier further study revealed that a link between the expression of reactive oxygen species including lipid peroxidation in conjunctival epithelia and the damage and inflammatory processes in conjunctiva of SS related dry eye 79.”
- On page 7, sub-heading 3.4
- statement about visual impairment and corneal complications among 1838 eyes makes no sense. What eyes are being referred to here?
I apologize that the explanation for eyes of primary and secondary SS patients is missing. We have added the explanation as follows.
Lines 382-383.
“A recent report show that severe visual impairment was noted in 10%, and 2.5% had corneal complications among the 1838 eyes of primary and secondary SS patients 81.”
- how does demonstration of the relevance of corneal nerve ablation in immune privilege support their “pathogenic” role in SS dry eye.
We appreciate the reviewer point out this issue.
The description and related reference is not related to pathogenic role in SS dry eye. Therefore, we have deleted this sentence.
“It is reported that corneal nerve ablation disrupts ocular immune privilege via attenuating CD103 on T regulatory cells.”
- On page 8, sub-heading 3.6
- what is “immune-mediated dry eye disease”? is this a different subset of dry eye disease?
We explained immune-mediated dry eye as dry eye disease with systemic immune mediated diseases as follows.
Lines 433-434
” In immune-mediated dry eye disease, including systemic autoimmune diseases such as SS, rheumatoid arthritis and systemic sclerosis and GVHD. “
- reference to “figure candidate” is unclear.
We apologize our mistake. We have deleted this phrase.
- Future directions are vague and refer to a generic need to clarify molecular mechanisms of SS-related dry eye disease development. Authors need to provide their specific views here with regards to autoimmune epithelitis that they intended to focus on but failed to do so effectively.
We appreciate the reviewer’s critical comments. We referred our hypotherical pathogenic process with a new Figure 2.
Lines 444-482.
4 Various components in causing Sjogrens’s syndrome related dry eye disease.
Based on the previous reports and findings, we propose the following model of pathogenic process in Sjogren’s syndrome (SS)-related dry eye disease. Possible pathophysiology of SS related dry eye disease may be initiated by autoimmune epithelitis in lacrimal gland, conjunctiva and meibomian gland epithelia under genetic predisposition and hormone imbalance. Viral infection is one of the candidates to initiate the abnormal expression of various proteins as SSA, SSB, M3 receptor, a-fodrin and Kallikrein. HLA-DR, adhesion molecules, and co-stimulatory molecules are expressed in conjunctival and lacrimal gland epithelia. Dysregulated autophagy play some role in presenting autoantigen and redistribution and translocation of protein related to tear component secretion molecules including b-fodrin, AQP5, Rab3D, and Angiotensin II type I receptor in lacrimal gland epithelia. CD4+ T cells are observed by attaching basal side of epithelia, suggesting that lacrimal gland and ocular surface epithelia present autoantigens to CD4+ T cells as non professional antigen presenting cells. On the other hand, it is suggested that CD8+ T cells also play some role to destroy the epithelia through perforin-granzyme pathway. In long term SS patients, accumulation of B cells and plasma cells in the stroma by producing excessive autoantibodies and accumulating abnormal autoantigens. (Figure 2)
Figure 2. Hypothetical shcema of pathophysiology of Sjogren’s syndrome related dry eye disease.
The one of the pathophysiology of Sjogren’s syndrome related dry eye disease may be initiated by autoimmune epithelitis in lacrimal gland, conjunctiva and possibly meibomian gland epithelia. Viral infection including Epstein Barr Virus (EBV), Hepatitis C virus (HCV), or Human T cell leukemia virus (HTLV) are possible candidates to initiate the abnormal expression of various autoantigen including Sjogren’s syndrome antigen type A (SSA), Sjogren’s syndrome antigen type B (SSB) , Muscarinic receptor type 3 (M3), or a-fodrin with HLA-DR, adhesion molecules, and co-stimulatory molecules in ocular surface and lacrimal gland epithelia.Translocation of various proteins related to tear component secretion may be influenced by dysregulated autophagy in SS. On the other hand, it is suggested that CD8+ T cells also play an important role to destroy the epithelia. In long term SS patients, accumulation of plasma cells in the stroma, leading to ocular supface epithelial dysfunction, resulting in severe dry eye disease. Proposed by Sumida T, et al with some modification.
- Overall, throughout the manuscript it is important to state clearly information known from experimental animal models and whether it has been corroborated in human SS patients. Stating observations from animal studies as conclusive findings in SS pathology is confusing and misleading.
I apologize the issues. I revised our manuscript whether the information from experimental animals or human SS patients.
Lines 121-123. the Kallikrein family of proteins, including Klk732, and Klk11 33 in humans and Klk13 34, and Klk22 35
Line 150. Autophagy was found to be a self-eating cellular process to maintain cellular homeostasis via lysosome-mediated degradation in yeast 40-42
Lines 156-158. Autophagy has been found to promote MH C class II presentation of peptides from intracellular source proteins in human B-lymphoblastoid Awells cell line in vitro 45.
Line 165. Enhanced autophagy and apoptosis are involved in the Ro/Sjogren’s syndrome (SS) A and La/SSB redistribution in secretory epithelial cells of the salivary gland in patients with SS 46.
Lines 184-186. It is confirmed abnormal trafficking of AQP5 contributes to the loss of secretory function in vitro in epithelial cells from SS patients. 50
Line 203. Rab3 D, a hallmark of mature secretory vesicles of epithelia, have also been translocated from the apical side to the basal side of lacrimal gland epithelia in SS patients 52,
Lines 215-218. In immune-mediated diseases, graft-versus-host disease in mice54, ATR2 is translocated from the apical membrane to the cytoplasm53, 54, similar to aquaporin and b-fodrin trafficking in lacrimal gland epithelia in SS patients.
Lines 235-236. In immune-mediated diseases, graft-versus-host disease in mice54, ATR2 is translocated from the apical membrane to the cytoplasm53, 54, similar to aquaporin and b-fodrin trafficking in lacrimal gland epithelia in SS patients.
Lines 260-261. ---glands than in those of their chronic GVHD-affected counterparts in patients62. In tissue specimens in patients with SS, CD4+ and CD8+ T cells are equally distributed in acinar
Lines 266-267. ---or development of pathology through affecting glandular epithelial cells in SS-related dry eye disease patients 63.
Lines 270-271. cytotoxic CD8+ T cells as SS gene signatures in whole blood and serum from patients with primary SS and further
Line 310. lacrimal gland ducts, lacrimal gland acini and lobules in mice71 72.
Line 311. Conjunctival epithelia in patients with SS
Line 316. The thickness of the membrane spanning mucin is reduced in SS patients compared to normal areas 73.
Lines 318-319. ----membrane spanning mucin in SS patients.
Line 343. ---to dysregulated conjunctival epithelia in patients with SS 73, 74.
Lines 345-346. the severity of dry eye disease by analyzing impression cytology specimensfrom patients with SS 83.
Lines 355-358. It is reported that IL-1b have anessential role in developing squamous metaplasia on the ocular surface epithelia in animal study 86 A significant role of IFN-g is extensively reviewed for human and animal study regarding mechanism of disease in SS by de Paiva, et al15,.
Line 379. a specific cytokeratin expression pattern in humans 90.
Lines 383. -----corneal complications among the 1838 eyes of primary and secondary SS patients 92.
Line 399. --in ocular surface in patients with SS 15, 100
Lines 436-438.
in patients with chronic GVHD with similar findings of dry eye disease in SS 21, 112 and in sclerodermatous mouse model111.
Submission Date
18 February 2021
Date of this review
02 Mar 2021 03:34:28

Round 2
Reviewer 1 Report
Concerns and suggestions have been sufficiently addressed.
Author Response
Thanks